# S100A4/FSP1: A Prognostic Marker and a Promising Target for Antitumor Therapy

**DOI:** 10.3390/ijms26199370

**Published:** 2025-09-25

**Authors:** Maria Bogachek, Alina Kazakova, David Sergeevichev, Sergey Vladimirov, Vladimir Richter, Anna Nushtaeva

**Affiliations:** 1Scientific Center of Genetics and Life Sciences, Sirius University of Science and Technology, 1 Olimpiysky Avenue, 354340 Sirius, Russia; maryambogachek@gmail.com (M.B.); kazakova.ala@talantiuspeh.ru (A.K.); sergeevichev.ds@talantiuspeh.ru (D.S.); vladimirov.sk@talantiuspeh.ru (S.V.); 2Institute of Chemical Biology and Fundamental Medicine, Siberian Branch of the Russian Academy of Sciences, Akad. Lavrentiev Ave. 8, 630090 Novosibirsk, Russia; richter@1bio.ru

**Keywords:** S100A4, FSP1, invasion, metastasis, fibrosis, tumor microenvironment, survival prognosis, RAGE, PROTAC, combination therapy

## Abstract

Numerous studies over three decades have confirmed the significant role of S100A4/FSP1 in the development of metastasis, the formation of the cellular and inflammatory components of the tumor microenvironment, and the development of fibrosis. S100A4 is a promising biomarker whose detection is associated with predicting overall survival in cancer patients. The action of S100A4 is mediated by extra- and intracellular signaling pathways involving targets currently used in the development of therapeutic agents, including monoclonal antibodies and drugs for targeted protein degradation. This review is devoted to the analysis of publications from the perspective of developing diagnostic predictive platforms and modern targeted antitumor therapy aimed at inhibiting the effects of S100A4, which allows avoiding the development of side effects and effectively modulates the tumor microenvironment to overcome immunosuppression and chemoresistance.

## 1. Introduction

Despite the success of modern therapy, cancer occupies a significant place in the structure of diseases leading to fatal outcomes: approximately one in five men or women develops cancer in a lifetime, whereas around one in nine men and one in 12 women die from it [1,2,3]. The expected number of new cases of cancer will be 35 million by 2050 [1]. An unfavorable disease outcome may be associated with a late diagnosis of an advanced-stage cancer and resistance to existing therapies. Thus, there remains an unmet healthcare need for the development of diagnostic systems that can detect early stages of cancer, as well as the development of new drugs to overcome drug resistance [4,5,6,7,8]. Criteria for finding a new target protein for therapy include its overexpression in tumors and tumor microenvironment compared to normal tissues, its correlation with patient survival, and its participation in tumor growth processes, metastasis, and immunomodulatory effects that transform a “cold” microenvironment into a “hot” one [9,10,11,12,13].

In this regard, of particular interest is the S100 family of calcium-binding proteins, which was discovered in 1965 and named after their concentration when dissolved in saturated ammonium sulfate. Many members of this family are involved in carcinogenesis, making them attractive targets for therapy and the development of prognostic markers [14]. Proteins of this family are known to influence cell signaling systems associated with cell division and differentiation, angiogenesis, apoptosis, and cell migration, as well as Ca^2+^-homeostasis and energy metabolism [14,15,16,17,18]. Currently, there are more than 20 known members of the S100 family, with about 50 percent homology, and their expression varies depending on the type of tissue [15].

One well-studied member of the S100 family is S100A4, also known as metastasin-1 (Mts1) or fibroblast-specific protein 1 (FSP1) [19], which was discovered by Ebradlize and colleagues [20]. Early in vivo studies have shown its association with metastatic progression [21,22]. Furthermore, more than 1500 publications over 25+ years confirm the role of S100A4 in the development of metastasis, chemoresistance, formation of the tumor microenvironment, as well as the promise of S100A4 as a new target for the therapy of treatment-resistant oncological diseases (Appendix A) [23].

The results of the studies showed that S100A4 is a prognostic marker for an unfavorable prognosis in the development of oncological diseases, including breast cancer, colorectal cancer, lymphoma, head and neck cancer, and others [23,24].

Tumors have been described as “fibrotic wounds that do not heal,” and chronic fibrosis has been identified as a risk factor for cancer development [25]. Recent observations also highlight the role of S100A4 in the development of fibrosis in oncological and inflammatory diseases by maintaining the desmoplastic response. This is indicative of tumor progression, with S100A4 playing a pivotal role in this process [26,27]. The protein S100A4 is expressed by cancer-associated fibroblasts (CAFs) and tumor cells. It has been demonstrated that S100A4 stimulates the activation of stromal fibroblasts and their hyperproduction of the extracellular matrix, leading to increased tissue stiffness. The resulting fibrotic microenvironment becomes a physical barrier that impedes the migration of immune cells into the tumor and the delivery of drugs. Furthermore, it promotes the invasion and metastasis of tumor cells and the development of drug resistance. Furthermore, the secreted CAF S100A4 is capable of paracrinally affecting cancer cells, enhancing their migration and invasion properties, thus closing the vicious circle of tumor progression and fibrosis that fuels the aggressiveness of the disease [28]. This makes S100A4 an essential component in the development of diagnostic systems and one of the most important markers of modern classifications of cancer-associated fibroblasts [23].

This review summarizes the extra- and intracellular mechanisms of S100A4 action and its expression in various tumor cell compartments and microenvironment, allowing us to suggest further trends of developing therapy and prognostic test systems for tumor diseases.

## 2. Structure of S100A4: Localization of the Gene and Regulation of Transcription of S100A4

The human *S100A4* gene is located in chromosome 1q21 and consists of two isoforms with three or four exons that encode a 101 amino acid residue protein [29,30]. The initial studies of transcriptional regulation date back to 1995, when Tulchinsky et al. found a correlation between high *MTS1* gene transcription in MOLT-4 lymphoma cells, human peripheral blood cells (macrophages, neutrophils, and lymphocytes), and DNA hypomethylation of the first exon and first intron of the *S100A4* gene (MTS1). Treating cells with a DNA methylase inhibitor increased S100A4 expression in low-expressing cell lines, confirming the importance of the epigenetic regulation of S100A4 transcription (MTS1) [29].

According to nuclear magnetic resonance and X-ray crystallography data, S100 protein monomers form a hydrophobic core and are capable of homo- and heterodimerization [31]. The monomer structure includes four α-helices and two loops of the EF-hand, which are Ca^2+^ binding helix-loop-helix motifs. It comprises a highly conserved C-terminal peptide comprising twelve amino acids. This is flanked by helices HIII and HIV and an N-terminal EF-hand of 14 amino acids, which differs from the consensus sequence and is itself flanked by helices HI and HII, thus forming a structure known as the S100-specific or pseudo-EF-hand [32].

The S100A4 protein structure is typical for the S100 family (Figure 1a), and its EF-hands contain a homodimer typical for S100 proteins. The intracellular and extracellular localizations of the S100A4 protein determine the signaling pathways of protein interaction involved in the implementation of cellular functions. One such function is the downstream cascade triggered by interaction with receptors for advanced glycation end products (RAGE) [31,33,34]. In the absence of Ca^2+^, the protein is in the apo form, adopting a conformation that allows interaction with other proteins. In order to carry out its regulatory function, the EF-hand must bind to Ca^2+^, followed by the initiation of conformational changes in the structure, and allows interaction with proteins in extra- and intracellular signaling pathways (Figure 1b) [14]. The signaling mechanism of extracellular S100A4 is poorly understood, but it is thought to involve the common S100 target RAGE, as well as other targets that have not yet been identified [34]. S100A4 secretion occurs in a non-canonical manner, possibly via lysosomal secretion, direct protein export, release of multivesicular bodies, or plasma membrane blebbing. In the external environment, S100A4 forms oligomers that can interact with several cell surface receptors (Figure 1c) [35].

The key signaling pathway regulating S100A4 expression is the Wnt/β-catenin signaling pathway [36,37,39]. The effect of this pathway on S100A4 transcription was described by Stein et al. using a colon cancer cellular model [39]. Specifically, it was revealed that S100A4 is a direct transcriptional target of the Wnt/β-catenin signaling pathway, and the binding site of the β-catenin/T-cell factor heterodimeric complex to the S100A4 promoter was mapped by electrophoretic mobility shift assays and chromatin immunoprecipitation analysis. Normally, β-catenin is degraded in the proteasome via a destruction complex, while in pathological conditions, β-catenin accumulates in the cytoplasm and moves to the nucleus. This leads to the activation of gene promoters and the expression of proteins responsible for the migration and invasion of tumor cells in vitro, and metastasis in vivo. Further studies of S100A4 transcriptional repression involved testing inhibitors of the Wnt/β-catenin signaling pathway. Several promising compounds with antitumor activity were identified in clinical trials, including Calcymycin and Niclosamide (Figure 2) [40]. In 2011, a high-throughput screen was performed to identify Calcimycin as a small-molecule targeting S100A4 promoter activity. Calcimycin’s mechanism of action involves inhibiting Wnt/β-catenin pathway activity and the expression of prominent β-catenin target genes, such as S100A4, cyclin D1, c-myc, and dickkopf-1.

In addition to the Wnt/β-catenin signaling pathway, there are other signaling pathways that influence S100A4 expression [41]. In 2003, Hernan et al. found that ERBB2 increased the expression of S100A4 in medulloblastoma cells via a pathway involving phosphatidylinositol 3-kinase, AKT1, and extracellular signal-regulated kinase 1/2 (Figure 1c) [41].

Therefore, detecting S100A4 protein expression could be a promising biomarker for cancer diagnostics and predicting its metastasis, suggesting the potential for developing S100A4 as a therapeutic target.

## 3. S100A4: Significance for Oncogenesis and Prognosis of Cancer

### 3.1. Expression and Prognostic Significance of S100A4 in Tumor Tissues

*Breast cancer.* According to most observations, S100A4 is an unfavorable prognostic marker for survival in breast cancer patients [42,43,44].

Several studies have been conducted to evaluate S100A4 expression in breast cancer samples and to correlate expression level with patient prognosis [42]. Rudland et al. evaluated the expression level and prognostic significance of S100A4 in patients diagnosed with breast cancer (349 patients, observed for 6 years, stage 1 and 2 breast cancer) using Western blot analysis. 41% of the samples were positive, showing visible staining of stroma and lymphocytes. They showed a correlation between S100A4 expression and shorter patient survival: 80% of negative patients survived after 19 years of follow-up, compared to 11% of positive patients (228 and 47 months, respectively) [42]. The prognostic value of S100A4 for survival in breast cancer patients was also confirmed by Platt-Higgins et al. in 2000 [44]. Immunocytochemistry confirmed S100A4 expression in 56% of patients. However, some researchers have not found S100A4 to have prognostic value. For example, in 2002, Pedersen et al. found a correlation between S100A4 expression in breast cancer biopsy specimens from 66 patients with no estrogen receptor expression and histological stage, but did not find a correlation with patient survival [45]. However, most subsequent studies have confirmed the association of S100A4 with survival prognosis.

In 1996, Grigorian et al. made an interesting observation [46]. They found that expression of MTS1 (S100A4) in MCF7 breast cancer cells resulted in tumor cells acquiring invasive growth properties and metastasizing in vivo. This was accompanied by a change in cell structure that manifested as a reduction in the number of desmosomes found at intercellular contacts, a hallmark of epithelial-mesenchymal transition [46]. Using in situ hybridization, S100A4 mRNA was localized in both the epithelium and stroma areas of breast tumors [47]. Their levels were higher in cancer tissues than in healthy tissues. Moreover, more aggressive tumors were characterized by elevated S100A4 mRNA levels compared to less aggressive ones. In 2006, Rudland et al. found a correlation between intense tumor staining for S100A4 and high vascularity. They also found a correlation between positive staining for S100A4 and decreased survival time in patients [43]. In 2016, Park et al. used immunohistochemistry to detect high levels of S100A4 expression in 524 invasive lobular breast carcinoma samples, which correlated with a high Ki-67 index and non-luminal type of breast cancer [48]. It should also be noted that in 2016, Zakaria et al. assessed S100A4 expression level in human breast cancer brain metastases in 138 patients and found 77% positive tumor cells staining of the nuclei and cytoplasm. Endothelial and smooth muscle cells were also stained [49]. Leu et al. provide a comprehensive account of the role of S100A4 in establishing the premetastatic niche of breast cancer cells in the lung, as well as the intricate signaling patterns between distinct cell types in the premetastatic lung. They also discuss the therapeutic potential of targeting S100A4 as a novel immunotherapy to suppress cancer metastasis at early and late stages of breast cancer progression [50].

*Endometrial cancer.* In 2009, Xie et al. showed that S100A4 plays a pivotal role in the invasion of endometrial cancer cells and is activated by the TGF-β1 signaling pathway [51]. In 2021, Ren et al. used the KLE endometrial cancer cells model to reveal that downregulation of S100A4 expression is linked to apoptosis induction and reduced proliferation [52]. In 2025, Wang et al. found that S100A4 expression was significantly upregulated in endometriotic lesions, playing a key role in regulating cell invasion and metabolic reprogramming in endometriosis. Suppression of S100A4 expression significantly decreased endometrial stromal cell migration and invasion, whereas overexpression of S100A4 abolished this inhibitory effect [53]. Also in 2025, Pei et al. demonstrated increased levels of S100A4 mRNA and protein in endometrial cancer samples compared to normal tissue, alongside an association between expression levels and tumor diameter, distant metastases, and lymph node metastases. S100A4 expression may be an independent risk factor for postoperative recurrence and the occurrence of metastases after surgery [54]. In the same year, S100A4 was validated as a promising biomarker and therapeutic target due to its involvement in metabolic reprogramming, which is important for the development of endometriosis. Overexpression of S100A4 in endometriosis lesions and its significant role in invasion were also demonstrated [55].

*Ovarian cancer.* In 2019, Link et al. studied S100A4 from 79 ovarian cancer patients by RT-qPCR and showed an excess of their level compared to normal samples [56]. This suggests that S100A4 transcripts could be potentially used as liquid biopsy markers [56]. Recent studies by Quiralte et al. (2024) on ovarian cancer-derived small extracellular vesicles using proteomic analysis have allowed us to conclude that S100A4 is associated with overall patient survival prognosis and resistance to platinum drugs [57]. In 2025, Hayashi et al. highlighted the high significance of S100A4 as an adverse prognostic marker for ovarian clear cell carcinoma (120 patients were studied by immunohistochemistry), causing tumor progression and chemoresistance triggered through anti-apoptotic mechanisms (increased BCL2:BAX ratio) and increased ALDH1high cancer stem cells levels [58]. In 2025, Ren et al. identified ALDH1A1 and S100A4 as possible genes associated with drug resistance using RNA-sequencing analyses. The study authors also note that ALDH1A1 and S100A4 expression levels were markedly elevated in platinum-resistant patient tissues compared to the sensitive cohort. A significant increase in S100A4 expression was observed in chemoresistant cells [59].

*Colorectal cancer.* Takenaga et al. employed Northern blot analysis to demonstrate that RNA levels of S100A4 were higher in colorectal adenocarcinoma samples than in normal mucosa RNA [60]. 94% of colorectal adenocarcinoma samples were positively stained by immunohistochemistry. In a similar study, Taylor et al. (2002) [61] used qPCR to detect increased S100A4 expression in colorectal cancer and liver metastases samples (n = 24) compared to normal intestinal tissue, as well as increased S100A4 expression in metastases compared to the initial intestinal carcinoma. Expression of S100A4 was also detected in T lymphocytes; thus, S100A4 may be involved in cytotoxic immune responses [61]. In 2013, Lee et al. evaluated 333 colorectal cancer samples by immunohistochemistry to assess the expression levels of E-cadherin, β-catenin, and S100A4. They found that, closer to the invasive tumor margin, there was a loss of E-cadherin and nuclear β-catenin, and a gain of S100A4. Expression of E-cadherin and S100A4, but not β-catenin, was a significant and independent marker for disease-free and overall survival [62]. Dahlmann et al. first demonstrated that high S100A4 expression in 60 primary tumors correlated with worse patient survival by studying the level of S100A4 mRNA in samples from 60 colorectal cancer patients [63,64]. Interestingly, the S100A4-interacting receptor RAGE was also overexpressed in patients with decreased overall survival as well as decreased metastasis-free survival, making RAGE an independent prognostic marker alongside S100A4 [63]. These findings suggest that S100A4 could be an attractive target for developing new agents against colorectal cancer [65]. In 2016, Boye et al. analyzed S100A4 expression using immunohistochemistry in colorectal cancer samples and found that nuclear S100A4 expression is a negative prognostic biomarker [66].

In 2024, Rasool et al. performed a study examining 80 patient samples using immunohistochemistry, qPCR, and western blot to establish the important role of S100A4 and S100A14 in colorectal cancer progression [67]. They detected a correlation between high S100A4 expression and invasion, advanced cancer stage, and metastasis development. Knockdown of S100A4 was found to decrease the migration, invasion, and proliferation of LoVo cells. Another study by Arif (2024) showed that high S100A4 expression is linked with tumor aggressiveness, whereas low S100A14 expression is associated with advanced disease stages and increased metastasis [68]. In 2025, Yan et al. showed that S100A4 is regulated by the miR-224-5p axis, which is involved in the development of 5-fluorouracil (5-Fu) resistance. This axis may therefore serve as a molecular marker for early prognosis and development of 5-FU resistance in patients in clinical practice [69].

*Gastric cancer.* In 2010, Wang et al. characterized 436 gastric cancer cases using RT-qPCR and found an association of S100A4 expression with the development of metastases and poor prognosis [70]. The value of S100A4 as a possible prognostic marker in gastric carcinomas was confirmed in 2013 by Zhao et al., who showed that S100A4 expression increased in the following order: gastritis, metaplasia, dysplasia, and gastric carcinoma [71]. Overexpression was also associated with greater invasion and metastasis, as well as tumor-node-metastasis staging [71]. In 2011, Feng et al. demonstrated a correlation between S100A4 and vascular endothelial growth factor C expression in gastric carcinoma (108 samples) and the development of lymph node metastases [72]. In 2014, Ling et al. conducted a meta-analysis of ten independent studies in Asia assessing the predictive value of S100A4 in relation to gastric cancer prognosis. They concluded that S100A4 overexpression significantly correlated with tumor grade, stage, metastasis, invasion, and relapse [73]. In 2022, Treese et al. investigated the role of S100A4 as a prognostic marker for adenocarcinoma of the stomach and esophagus. They examined tissue microarray samples from 277 patients and showed that high S100A4 expression is associated with lower survival rates and the development of metastases [74]. In 2025, Tang et al. demonstrated that increased S100A4 activity via the OSTM1-S100A4 axis promotes gastric cancer progression by altering the gastric tumor microenvironment through increased angiogenesis and fibroblast activation [75].

*Liver cancer.* In 2021, Sun et al. performed a study investigating the functional role of exosomes in hepatocellular carcinoma and showed in vitro and in vivo that S100A4-rich exosomes are drivers of metastases [76]. The mechanism of action of S100A4 was found to be associated with the induction of osteopontin expression, a known stemness-related protein, via STAT3 phosphorylation. They also demonstrated that high levels of S100A4-rich exosomes in patient plasma are associated with a poor prognosis and may serve as a prognostic marker for the development of metastases in hepatocellular carcinoma. In 2025, Qin et al. confirmed earlier findings that S100A4 promotes HCC cell migration, invasion, and metastasis by activating the EMT process via NMIIA, possibly via exosome-mediated signaling [77].

*Pancreatic cancer.* In 2013, Tsukamoto showed an important role of S100A4 in pancreatic cancer invasiveness [78]. In the same year, Kozono revealed that S100A4 mRNA expression could predict pancreatic cancer cell radioresistance, suggesting its potential role in poor response of pancreatic cancer cells to radiotherapy [79]. In 2019, Jia et al. recommended using S100A4 expression alongside preoperative serum CA19.9 levels as dual prognostic biomarkers for pancreatic cancer prognosis, based on a study of immunohistochemical staining for S100A4 in 128 PC [80].

*Renal cancer.* In 2012, Wang et al. compared the levels of proteins and mRNA in renal epithelial neoplasms, including 155 clear cell renal cell carcinomas, and found higher S100A4 expression in glomerular epithelium and endothelium, distal tubules and collecting ducts, and loops of Henle compared with normal tissue. S100A4 expression level was confirmed to be a poor prognostic factor [81]. Wen et al. suggest that manipulating S100A4 may provide a beneficial therapeutic strategy for chronic kidney disease [82].

*Thyroid cancer.* In 2005, Zou et al. demonstrated that S100A4 overexpression is associated with thyroid tumor invasion and metastasis, suggesting its potential as a therapeutic target [83]. In 2016, Zhang showed that S100A4 knockdown blocks the growth and metastasis of anaplastic thyroid cancer cells in vitro and in vivo [84]. Furthermore, in 2024, it was reported that the presence of S100A4 in tear fluid can serve as a biomarker for the predisposition to thyroid-ophthalmological pathology in patients with autoimmune thyroiditis [85].

*Eye cancer.* S100A4 has been found to be expressed in various types of eye cancer, including melanoma and other tumors [86,87,88]. S100A4 has been implicated in eye tumor metastasis formation [89,90]. Chen et al. showed that S100A4 is a biomarker of uveal melanoma, and its high expression is related to poor prognosis [91]. In 2025, Tao et al. demonstrated significantly higher S100A4 expression in the high-risk group of uveal melanoma patients compared to the low-risk group, and moreover, they found that this was associated with a decreased survival rate [92].

*Head and neck cancer.* In 2013, Rasanen et al. revealed a 20-fold predominance of S100A4 expression in mesenchymal cells using proteomic analysis of the secretomes of E-cadherin high epithelial-like and E-cadherin low mesenchymal-like head and neck squamous cell carcinoma samples [93]. They also found that, in an organoid cellular model, less invasion occurred with reduced S100A4 expression, making S100A4 a promising therapeutic target. In 2022, Gao et al. showed that a poor prognosis in patients with head and neck squamous cell carcinoma is associated with high mRNA levels of S100A10, whereas high mRNA levels of S100A3, S100A4, S100A7A, and S100A9 are associated with high patient survival [94]. Thus, in some cases, other members of the S100 protein family can serve as predictive markers for patient survival and functions such as invasion, angiogenesis, EMT, and hypoxia, as verified by S100A10 single-cell RNA sequencing analysis.

*Brain cancer.* S100A4 expression was evidenced as higher in glioblastoma, compared with low-grade astrocytic tumors, suggesting its involvement in glioma progression [95]. In 2017, Chow et al. established S100A4 as a central node in a molecular network that controls stemness and epithelial-mesenchymal transition in glioblastoma, suggesting S100A4 as a candidate therapeutic target [96]. Inukai et al. (2022) demonstrated that knockdown of S100A4 in the glioblastoma cell line KS-1 decreased migration capability, concomitant with decreased Slug expression [97]. In 2025, Wong et al. indicated that S100A4 is a promising therapeutic target [98]. They used a bispecific S100A4/TFR antibody to reprogram the glioblastoma tumor microenvironment and target glioblastoma stem cells.

*Lung cancer.* In 2000, Kimura et al. used immunohistochemical detection to analyze S100A4 expression in 135 non-small cell lung cancer (NSCLC) samples. S100A4 expression was detected in 60% of cases and correlated with a poor prognosis for patients [99]. A meta-analysis conducted in 2020 found that increased S100A4 overexpression is an unfavorable prognostic marker and is associated with tumor progression in NSCLC patients. In 2023, Kagimoto showed that the accumulation of serum S100A4 and [18F]-fluoro-2-deoxy-D-glucose in the non-cancerous region of benign interstitial pneumonia significantly predicts lung cancer after lung resection, potentially aiding the selection of a treatment strategy [100]. Also in 2022, Wu et al. undertook a series of experiments to elucidate the role of exosome-transmitted S100A4 in the progression of NSCLC. They showed that S100A4 knockdown is associated with decreased proliferation, migration, and invasion and increased apoptosis of NSCLC cells. Interestingly, S100A4 knockdown negatively affected PD-L1 expression, and this effect was neutralized by STAT3 overexpression. This confirmed the importance of activation of this signaling pathway to implement S100A4 functions. Furthermore, the immunosuppressive role of S100A4 on the T cell component was demonstrated through STAT3 activation [101]. In 2022, Deo et al. showed in vitro that interaction with lung fibroblasts induces the secretion of S100A4 platinum-taxol-resistant A2780 metastatic epithelial ovarian cancer cells. This interaction occurs via the insulin-like growth factor 1 receptor (IGF1R)-α6 integrin-S100A4 molecular network, which mediates the development of metastases and chemoresistance. The authors therefore propose that S100A4 may be a potential therapeutic target for metastatic ovarian cancer [102]. This observation corroborates an earlier study by Hoshino et al. (2015), which found that metastatic lung cancer cells released α6β4—expressing exosomes that fuse with the fibroblast cells, triggering S100A4 upregulation and enhancing the development of lung metastases [103].

While the S100 protein family plays a significant role in the pathogenesis of NSCLC, its role in small cell lung cancer (SCLC) has received less attention [99]. In an extensive review, Wang T. and Liu R. analyzed the involvement of S100 proteins in lung cancer from 2021 to 2025. They noted that, for NSCLC, the S100A2, S100A6, S100A7, S100P, and S100A4 proteins are often associated with tumor progression. Furthermore, S100A4 is the only potential oncogenic factor in SCLC [17].

Thus, it is clear that S100A4 remains a key research target in oncology in 2025. Its role in regulating cell growth, apoptosis, and metastasis is the subject of ongoing research, particularly in relation to aggressive tumors. Additionally, studies are underway to assess its potential as a biomarker for early diagnosis and prognosis of the disease, as well as a therapeutic target in the development of new anti-tumor strategies. Further research in this direction could lead to new opportunities for personalized treatment of oncological diseases (see Appendix A).

### 3.2. Signaling Pathways That Mediate S100A4 Functions in Tumors and Microenvironment

*S100A4/RAGE signaling.* The involvement of signaling pathways in the implementation of S100A4 functions depends on the protein’s extra- or intracellular localization. One of the main receptors that interacts with the extracellular form is RAGE (Receptor for Advanced Glycation Endproducts), which is known to be an important ligand-binding node that triggers signaling pathways associated with carcinogenesis, tumor propagation, and metastatic process [33,104,105,106].

In 2014, Dahlmann et al. first demonstrated that the interaction between extracellular S100A4 and RAGE activates the MAPK/ERK and hypoxia signaling pathways, thereby increasing the invasion and migration of colorectal cancer cells [63]. S100A4 and RAGE binding also triggers a pathway that regulates tumor cell migration. It is an alternative to the well-known intracellular mechanism whereby S100A4 binds to non-muscle myosin. In 2015, Medapati et al. showed that extracellular S100A4 binds to RAGE, activating a signaling cascade that depends on the intracellular partner Dia-1 (diaphanous-1), resulting in the activation of Cdc42 and RhoA small GTPases [107]. They also demonstrated the possibility of alternative activation of ERK signaling by extracellular S100A4 outside the RAGE-related signaling pathway, which does not determine the migration of tumor cells. Thus, the authors propose targeting the RAGE/Dia-1/small GTPases signaling to overcome metastatic forms of thyroid cancer. Further studies confirmed the significance of S100A4/RAGE interaction in other oncological diseases.

In 2016, Herwig et al. described the secretion of S100A4 by A375 melanoma cells. Extracellular S100A4 interacts with RAGE receptors and promotes prometastatic activation of A375 cells [108]. In 2019, Ryan et al. investigated the metastatic process of triple-negative breast cancer cells overexpressing geminin and established the criticality of S100A4/RAGE signaling for vascularization and the development of invasion due to maintenance of stemness and epithelial-to-mesenchymal phenotype [109]. In 2020, Wu, Y. et al. showed that the stimulation of airway smooth muscle tissues with IL-13 or TNF-α promotes secretion of the extracellular form of S100A4 and expression of RAGE and S100A4 [110]. Stimulation of NF-κB or Akt signaling in tissues has also been shown to be mediated through the interaction of S100A4 with RAGE receptors. The study by Kim et al. (2021) demonstrated that S100A4 released from aggressive prostate cancer cells stimulated osteoclast development via the cell surface receptor RAGE. This suggests that S100A4/RAGE plays a key role in forming a niche for bone metastases [111].

New studies by Park et al. (2023) reveal a S1004/RAGE-mediated universal mechanism of metastasis for breast cancer cells, bladder, and lung through Padi4-mediated nuclear expiration in tumor cell destruction and apoptosis [112]. The extracellular DNA-protein complex contains a large number of chromatin-bound RAGE ligands, including S100A4. Activation of RAGE receptors on non-disrupted surviving tumor cells by S100A4 leads to Erk activation and stimulates proliferation and metastasis.

Thus, the accumulated data support the key role of the S100A4/RAGE signaling pathway in triggering and regulating tumor growth and metastasis, making it a promising target for therapy (Figure 3).

*Annexin A2-S100A4 complex.* Annexins and S100 proteins are two large families of calcium-binding proteins that are known for their ability to form functional complexes [113,114].

Annexin A2 (ANXA2), which is structurally associated with the cytoskeleton and cell membrane, participates in intracellular vesicle fusion, the organization of membrane domains, lipid rafts, and membrane-cytoskeleton contacts. It also participates in extracellular processes such as plasmin regulation and cell-cell interactions. ANXA2 is overexpressed in tumors and promotes tumorigenesis and cancer progression [115,116]. ANXA2 forms complexes with S100A4 and S100A10 heterotetramers that differ in structure: the ANXA2-S100A10 complex is formed when 2–14 amino acids of ANXA2 bind to S100A10. Whereas, the entire internally disordered N-terminal domain (2–33 amino acids) of ANXA2 wraps around the S100A4 domain in the ANXA2-S100A4 complex [117].

Semov et al. (2005) showed colocalization and direct binding of the extracellular form of S100A4 and the N-terminal region of annexin II, which is expressed by endothelial cells [86]. This leads to plasminogen activation and the digestion of tissue barriers, thereby potentiating metastasis. Interaction with annexins is a characteristic feature of many members of the S100 family and is associated with tumor-induced angiogenesis and metastasis. In 2020, Ecsedi et al., using ANXA2 as a heterologous fusion partner, were able to solve the atomic resolution structure of a challenging crystallization target, the transactivation domain (TAD) of p53 in complex with the metastasis-associated protein S100A4 [118]. p53 TAD forms an asymmetric fuzzy complex with the symmetric S1004 and could interfere with its function. Therefore, blocking this interaction makes ANXA2-S100A4 an attractive target for new anti-cancer therapies.

### 3.3. Intracellular Signaling Pathways for S100A4 Functions

*NMIIA (the nonmuscle myosin 2A).* According to a number of publications, S100A4 plays a critical role in the development of metastases by interacting with NMIIA (the nonmuscle myosin 2A) [119]. Filament assembly of NMIIA is selectively regulated by the small Ca^2+^-binding protein, S100A4, which causes enhanced cell migration and metastasis in certain cancers [120].

Initial studies by Kriajevska et al. in 1994 showed an association between the MTS-1 (S100A4) protein and the heavy chain of non-muscle myosin using the blot overlay technique and immunoprecipitation [119]. Co-localization of S100A4 and the myosin complex was determined by immunofluorescent staining of the mouse mammary adenocarcinoma cell line. Increased migration of mouse mammary adenocarcinoma cells mediated by S100A4 (Mts1) occurs through interaction with tropomyosin and non-muscle myosin II in a Ca^2+^-dependent manner, which inhibits the MgATPase activity of myosin in vitro [121].

The overexpression of S100A4 induces metastasis by increasing cytoskeletal plasticity and migration [122]. The marginal localization of S100A4 and the formation of protrusions by migrating cells were demonstrated. In 2003, Li et al. mapped the S100A4 binding site to 1909–1924 amino acids of the myosin-IIA heavy chain C-terminal tail of the alpha-helical coiled-coil [123]. These 16 amino acid binding sites are typical of S100 family members that regulate myosin IIA assembly through monomer sequencing. Kiss et al. and Ramagopal et al. elucidated the crystal structure of the S100A4-NM IIA complex [124,125]. Thus, the S100A4 is able to regulate cell migration and shape [125,126,127,128].

Further studies have revealed the significant roles played by S100A4 and NMIIA in tumor progression. In 2020, Tochimoto et al. demonstrated that increased promoter activity and S100A4 mRNA expression, initiated by NF-κB/p65 transfection, were responsible for the activation of the endometrial carcinoma cell line model, thus confirming the regulatory function of NF-κB/p65 [129]. The relationship between S100A4 overexpression, increased tumor cell migration, and cancer stem cell enrichment has been shown. The interaction between S100A4 and NMII heavy chains, including myosin 9 and myosin 14, has been mapped. Specific inhibition of NMII by blebbistatin phenocopied S100A4 overexpression and induced a fibroblast-like morphology. In 2020, Hiruta et al. reported on the significance of the S100A4/NMIIA/p53 axis in the development of epithelial-mesenchymal transition and cancer stem cell markers using a model of high-grade serous ovarian carcinoma [130]. S100A4 knockdown also led to apoptosis and reduced cell proliferation [130]. In 2022, Inukai et al. showed the importance of the S100A4/NMIIA axis in enhancing migratory capability, promoting hypoxia and vascularization in the glioblastoma microenvironment [97].

Thus, the totality of the data indicates the important role of the S100A4/NMIIA interaction in regulating migration, maintaining tumor cells and epithelial-mesenchymal transition, and forming a pro-metastatic microenvironment. Blocking S100A4/NMIIA binding may therefore represent a promising strategy to suppress metastases.

*Wnt/β-catenin signaling*. Wnt/β-catenin signaling is an important component in tumor development [131,132,133,134]. In the absence of Wnt binding, cytoplasmic β-catenin is phosphorylated by a destruction complex, resulting in its ubiquitination and subsequent degradation by the proteasome. In the presence of Wnt binding, an increase in intracellular β-catenin is observed, with its transition to the nucleus where it accumulates and interacts with T-cell-specific factor/lymphoid-enhancer-binding factor and co-activators, initiating transcription of Wnt target genes. In 2006, Stein et al. demonstrated that S100A4 expression is regulated by the β-catenin/T-cell factor in colorectal cancer cell line models, leading to increased migration and invasion [39]. In 2016, a detailed review was published on the regulation of S100A4 activity by the canonical β-catenin-dependent Wnt pathway [65]. Activation occurs when Wnt binds to the Frizzled receptor and the co-receivers, low-density lipoprotein receptor-related proteins (LRP) 5/6, which bind to Wnt. This renders the destructive β-catenin complex inactive, allowing β-catenin to accumulate in the cytoplasm and move into the nucleus. There, it activates the transcription of S100A4 and other proteins via a T-cell factor binding motif along and other factors [135]. In 2024, Yang et al. showed that the S100A4 regulates the proliferation and apoptosis of benign prostatic hyperplasia through the ERK pathway, as well as modulating fibrosis via Wnt/β-catenin signaling [136].

Interestingly, the authors demonstrated using both in vitro and in vivo models that S100A4 knockdown decreases benign prostatic hyperplasia cell proliferation and fibrosis markers. This suggests that S100A4 could be a viable therapeutic target.

*The role of metalloproteases.* One of the main factors contributing to the development of metastasis in tumor diseases through extracellular matrix degradation is the activity of matrix metalloproteases (MMP) [137]. The up-regulation of MMP-2 and MMP-9 has been demonstrated in the case of S100A4 overexpression, which allows cell invasion and promotes metastasis [138,139]. In this study, the authors demonstrated that S100A4 binds to RAGE, initiating a signaling cascade that ultimately leads to the release of MMP-13 via the nuclear kappa light chain factor of activated B cells (NF-KB) pathway in articular chondrocytes [140]. High S100A4 and MMP12 expression was significantly correlated with poor prognosis in patients with uveal melanoma [91]. Blocking S100A4/MMPs interactions may help to prevent metastasis.

*FGF2/FGFR1.* The fibroblast growth factor (FGF)/fibroblast growth factor receptor(FGFR1) is known to affect tumor progression and metastasis, as measured by its effects on angiogenesis and the stromal component of the microenvironment [141]. In 2021, Santolla et al. showed that FGF2 increased S100A4 secretion via FGFR1, along with the ERK1/2-AKT-c-Rel signaling transduction in cells of triple-negative breast cancer [142]. In vitro studies have also confirmed that activation of the S100A4/RAGE pathway promotes angiogenesis and stimulates the migration of cancer-associated fibroblasts (CAFs). Therefore, preventing the interaction between S100A4 and FGF2 may modulate the migration of CAFs, thereby promoting angiogenesis.

*P53.* p53 is a well-studied tumor suppressor [143,144]. Mutations and loss of functionality have been shown to lead to tumor progression. Currently, the protein itself is considered undruggable, and strategies aimed at finding and blocking the cofactors that cause degradation and weaken the tumor suppressor function of p53.

Several in vitro studies have confirmed the interaction of S100A4 and p53, and there are data suggesting that S100A4 can degrade p53, resulting in decreased cisplatin-induced apoptosis and increased tumor formation [130,145,146]. In the nucleus, S100A4 binds to the C-terminal transactivation domain of p53 to modulate its pro-apoptotic function and promotes p53 degradation [145]. A recent study by Harada et al. (2024) showed that S100A4-mediated inhibition of the p53/p21(WAF1) axis leads to increased proliferation of locally advanced rectal carcinoma, noting an association with the development of adriamycin-induced apoptosis chemoresistance [147]. 

Inhibiting S100A4 can reduce degradation and p53 suppression, thereby potentiating tumor cell apoptosis and helping to overcome chemoresistance.

*TGFβ1/SMAD/S100A4 signaling.* TGFβ1 is a key protein in cancer progression, promoting invasion, stemness, immunosuppression in the tumor microenvironment, and therapeutic resistance [148,149]. Upon activation of fibroblasts, S100A4 promotes TGF-β1-induced profibrogenic function [150]. Treatment of endometrial TGFβ1 cancer cells in vitro has been shown to promote S100A4 expression, which plays a critical role in increasing the motility and invasiveness of endometrial cancer cells. Thus, in invasive endometrial cancer, the TGFβ1/SMAD/S100A4 axis is a significant regulator of invasive potential, and TGFβ1 blockers may be a promising therapeutic for inhibiting that pathway [51,151].

*MTA1-S100A4-NMIIA: Role in angiogenesis.* Angiogenesis, the formation of new blood vessels, is an important process supporting tumor growth, invasion, and metastasis [152]. In 2019, Ishikawa et al. demonstrated the colocalization of S100A4 and another well-studied, pro-invasive protein, Metastasis-Associated 1 (MTA1), in mouse endothelial cells [153]. MTA1 maintained stability and regulated S100A4 expression. Both proteins control NMIIA phosphorylation. In the human prostate cancer xenograft model, it was demonstrated that S100A4 knockdown reduces tumor growth due to suppression of angiogenesis. These data confirm the previously obtained data and indicate the role of S100A4 in the development of tumor vascularization [154]. In 2005, Ambartsumian et al. observed the development of hemangiomas in a transgenic mouse model with ubiquitous S100A4 expression, which proved the protein’s involvement in angiogenesis in vivo [154].

Thus, S100A4 and MTA1 form a functional axis (MTA1-S100A4-NMIIA) that promotes tumor progression and is of interest in the development of anti-angiogenic therapeutics.

*The role of S100A4 in epithelial-mesenchymal transition (EMT).* Epithelial-mesenchymal transition (EMT) in tumor cells is characterized by the acquisition of mesenchymal markers and decreased expression of epithelial markers, including E-cadherin [155,156,157]. A distinctive feature of EMT is the enrichment of the tumor with a subpopulation of cancer stem cells (CSC), which are responsible for the development of metastases and for chemoresistance and radioresistance. The role of S100A4 as a driver of EMT in different tumor types has been well studied, with several authors demonstrating the involvement of various signaling systems in EMT, including the Sonic hedgehog-Gli1 (Shh-Gli1) axis and the inhibitor of nuclear factor kappa B NF-κB kinase (IKK)/nuclear factor kappa B (NF-κB) pathway [158,159]. In 2014, Xu et al. showed the importance of the Shh-Gli1 signaling pathway in the upregulation of the S100A4 and vimentin genes, and the downregulation of E-cadherin in a pancreatic cancer cell model [158]. In 2014, Sung et al. revealed that loss of membranous expression of E-cadherin and β-catenin expression, along with increased S100A4 expression, were more prevalent in the invasive margin than in the tumor center of Vater cancers, demonstrating the importance of EMT for invasion [160]. In 2014, Wang et al. demonstrated that TGFbeta initiates the activation of the LIM and SH3 protein 1 (LASP1)/S100A4/Smad pathway. S100A4 was upregulated by LASP1, leading to EMT transitions and increased invasion of colorectal cancer cells [161]. Zhu et al. (2018) identified high levels of S100A4 expression in the CSC bladder [159]. Through immunoprecipitation, they detected a direct S100A4—IKK binding, which affects the IKK/NF-κB signaling pathway, and activates NF-κB and its downstream targets, including TNFalpha/IL2, thereby increasing bladder CSC proliferation. In 2017, Chow et al. showed that S100A4 is a key protein that determines glioma stemness, exercising its function by influencing transcription factors that are important for triggering EMT, including SNAIL2 and ZEB [96]. In 2016, Hua et al. concluded, when studying endometrial cancer, that S100A4 was associated with disease progression, cell migration, and invasion of cells through EMT regulation [162]. In 2016, Xu et al. identified the mechanism by which S100A4 affects the stemness of triple-negative breast cancer in vitro via MMP2 regulation [163]. In 2020, Li et al. demonstrated the effect of S100A4 on tumorigenesis and the development of fibrosis in the liver, as well as its impact on stemness, in an S100A4-deficient mouse model [164]. This effect was found to be associated with synergy with the extracellular matrix component collagen I and dependent on RAGE/beta-catenin signaling. In 2021, Kim et al. discovered that intracellular S100A4 plays an important role in EMT in prostate cancer, while secreted S100A4 mediates the development of a metastatic process in bones [111]. 

Thus, S100A4 is an attractive target for developing therapies that target cancer stem cells and block EMT (see Appendix A).

*S100A4 and regulation of anaerobic glycolysis in hypoxia.* Hypoxia is a tumor hallmark, leading to increased metastasis and chemoresistance of tumor cells that use glycolysis as the principal mode of energy supply—a phenomenon referred to as the «Warburg effect» [165,166,167]. In 2023, Li et al. described hypoxia-dependent S100A4 expression in lymphatic endothelial cells (LECs) [168]. Subsequent studies employing knockout mice in the whole body or specifically in LECs showed the criticality of this expression for tumor lymphangiogenesis and the process of metastasis to lymph nodes. The motility of tip cells and the activation of AMPK-dependent glycolysis during lymphatic sprouting were found to be contingent on S100A4.

Consequently, the inhibition of S100A4 has the potential to disrupt the energy supply, leading to the demise of tumor cells and preventing their adaptation to hypoxia through the «Warburg effect».

*The tumor microenvironment: study of S100A4 expression and use for classification of CAFs.* The effect of the tumor microenvironment, including the stromal component, on disease prognosis and response to therapy has been the subject of extensive research [169,170,171]. Numerous studies have shown that the protein S100A4 is expressed at a higher level in stromal tissue compared to epithelial tissue [172]. In addition to fibroblasts, a variety of cells within the microenvironment, including macrophages, T lymphocytes, CAFs, and MDSCs, have been observed to express S100A4 [173]. Through various signaling pathways, these cells contribute to the formation of a pro-inflammatory microenvironment and matrix remodeling (Figure 4). This was recently reviewed by colleagues in 2025 [23]. Initial studies conducted in 1995 by Strutz et al. showed that the color on the FSP1 stained the cytoplasm of fibroblasts, but not epithelium [174]. Further in vitro studies have shown that S100A4 is associated with the transition from epithelium to fibroblasts, as evidenced by the fact that overexpressing FSP1 cDNA in renal tubular epithelium led to EMT. Murine models have shown that FSP1 is expressed preferentially in the stroma during embryogenesis and in the interstitial stroma of adult mice, including the skin, prostate, and kidneys. The source of FSP1 (+) fibroblasts is migration to normal interstitial spaces from bone marrow and also local EMT during renal fibrogenesis [175].

The FSP1 expression scores CAFs as a heterogeneous population consisting of several functional subtypes, which are distinguished by molecular markers. Various approaches have been used to classify CAFs, including morphological studies, analysis of cytometry data and transcriptomic analysis data, single-cell RNA-sequencing, and the most up-to-date approach, which uses artificial intelligence [176,177]. S100A4 expression evaluation is also used to evaluate CAF reprogramming in mesenchymal-epithelial transition induced by hypoxia [178]. Currently, CAF typing involves evaluating the levels of several molecular markers, including vimentin (Vim), smooth muscle alpha-actin (SMAα), fibroblast activation protein (FAPα), platelet-derived growth factor receptor (PDGFRα), and others, along with fibroblast-specific protein 1 (FSP1/S100A4) [176]. In 2004, Bhowmick et al. demonstrated that TGF-β1 activates resident fibroblasts in CAFs that express alpha-smooth muscle actin (αSMA, ACTA2), periostin (POSTN), α-fibroblast activation protein (αFAP, a.k.a. dipeptidylpeptidase IV), and fibroblast specific protein-1 (FSP-1, a.k.a. S100A4), and produce type I collagen [179]. In 2006, Sugimoto et al. studied murine breast—and pancreatic CAF populations and found that S100A4/FSP1 identifies a unique population of fibroblasts [180]. They also emphasized the importance of using S100A4 in CAF typing [180]. In an earlier study in 2011, O’Connell showed that S100A4-positive fibroblasts express tenascin-C and VEGF-A, which supports the formation of metastases [181]. In 2013, Rasanen et al. performed a comparative secretome analysis of epithelial and mesenchymal subpopulations of head and neck squamous cell carcinoma and identified a 20-fold predominance of S100A4 expression in the mesenchymal cells compared to epithelial cells [93]. The mechanism of the effect of S100A4 on metastasis and invasion was associated with increased MMP2 expression. In 2018, Liu et al. demonstrated that ATP stimulates the intracellular expression and secretion of S100A4, contributing to an increase in breast cancer cell motility and the transformation of fibroblasts into CAFs [182]. Niclosamide has been shown in vivo to inhibit the development of metastases and to attenuate the effect of CAFs on tumor cells, which promotes their migration. In 2017, Zhang et al. studied the stroma of intrahepatic cholangiocarcinoma using immunohistochemistry and determined a high percentage of CAF positivity for S100A4 staining (84.5%) [183]. They also found an association between S100A4 expression and lymph node metastasis, as well as an immature phenotype characterized by an advanced TNM stage and poor five-year overall survival. In 2018, Jiao et al. demonstrated in vivo the possibility of reducing stemness and inflammatory processes in hepatocellular carcinoma by depleting S100A4+ stromal cells [184].

In 2018, Costa et al. identified four subtypes of CAF (S1–S4) in the tumor microenvironment of breast cancer. They based their identification on flow cytometry data and the expression of FSP1/S100A4, FAPα, integrin β 1 (CD29), α-SMA, platelet-derived growth factor receptor β (PDGFR β), and caveolin-1 [185]. Two of the subtypes, CAF-S1 and CAF-S4, express αSMA and can be considered myofibroblasts. A distinctive feature of the CAF-S1 subgroup is its high level of expression of FAPα, associated with adhesion, wound healing, and immunosuppression. CAF-S4s are characterized by the FAP^low^/αSMA^high^ phenotype associated with invasion and metastasis [186,187]. Due to smooth muscle alpha-actin (SMAα) expression, CAF-S1 and CAF-S4 subtypes are myofibroblasts. The isolation of CAF-S3 into a separate subtype is based on the expression of fibroblast-specific protein 1 (FSP1/S100A4) and platelet growth factor receptor α (PDGFRα). CAF-S3 fibroblasts (CD29^Med^ FAP^Neg^FSP1^Med-High^α-SMA^Neg^ PDGFRβ^Med^ CAV1^Low^) are also observed in healthy tissues. Subtypes CAF-S2 and S3 are considered non-activated fibroblasts with functions that are not fully understood [173,188,189]. Using flow cytometry, immunohistochemistry, and RNA sequencing, Pelon et al. showed the predominance of the myofibroblastic subtypes of CAF-S1 and CAF-S4 in breast cancer metastases [187]. Thus, S100A4 is a successful marker for classifying CAFs and identifying CAF subtypes associated with metastasis development. However, the significance of the CAF-S3 subtype with enriched S100A4 expression remains unclear, and further studies are needed to understand the potential activation of this CAF subtype.

In 2018, Givel et al. used the proposed CAF-S1-to-S4 classification for BC to identify four subsets of CAFs in high-grade serous ovarian cancers (HGSOC) and determine the CXCL12β-associated stromal heterogeneity and immunosuppressive environment in mesenchymal HGSOC [190]. In 2020, Gil Friedman and his colleagues used single-cell RNA-seq to isolate two major CAF populations in mouse samples [191]. They termed these populations pCAF and sCAF, based on the selective expression of the markers Pdpn or S100A4. Subsequently, S100A4-fibroblasts were successively divided into two subsets of CAFs: the Spp1^high^S100A4^low^ subset was enriched for signatures of antigen presentation (H2-Aa) and ECM remodeling, while the Spp1^low^S100A4^high^ subset was enriched for protein folding and metabolic genes. Normal mammary fat pads were dominated by Pdpn+ fibroblasts, whereas primary tumors had more S100A4+ fibroblasts. The MHC class II cell-surface receptor HLA-DR marked a subset of sCAFs. When comparing two groups of patients with different BRCA1/2 mutational statuses, higher S100A4/PDPN ratios were observed in patients with mutations compared to those with wild-type BRCA.

*The tumor microenvironment: mesenchymal stem cells.* In 2019, Ryan et al. investigated the metastatic process of geminin-overexpressing triple-negative breast cancer cells and established the importance of S100A4/RAGE signaling for vascularization and invasion due to their stemness and epithelial-to-mesenchymal phenotypes [109]. The detailed mechanism involved the secretion of acetylated HMGB1 (Ac-HMGB1) by tumor cells, followed by activation of RAGE and CXCR4 expression on mesenchymal stem cells (MSCs) located in the tumor stroma. Further, the grandiose transformation of the tumor microenvironment, including (1) the differentiation of MSCs into S100A4-secreting CAFs, (2) the secretion of CCL2 by CAFs, (3) the attraction of M0-macrophages from the stroma into the tumor under the influence of CCL2, and (4) the CCL2 mediated polarization of M0-macrophages into Gas6-secreting M2-tumor-associated macrophages (M2-TAMs) was revealed. In 2023, Shufeng Ji et al. investigated the mechanisms of immunosuppression in the microenvironment of triple-negative breast tumor microenvironment and found that S100A4 secreted CTLA4+ T cells, contributing to the stem phenotype cells [28]. Grum-Schwensen et al. demonstrated the efficacy of a 6B12 S100A4-neutralizing antibody in a spontaneous breast cancer model. The antibody’s mechanism of action was based on preventing T cells from adhering to the tumor and metastatic niche, as well as shifting the Th1/Th2 balance toward the pro-tumor phenotype Th2 [192].

*The tumor microenvironment: macrophages.* In 2022, Qi et al. identified the effect of a macrophage-secreted form of M2 S100A4 on extracellular matrix remodeling and fibroblast activation [193]. This creates conditions that allow for the formation of a premetastatic niche via the extracellular signal-regulated kinase (ERK) signaling pathway. In 2022, Liu et al. demonstrated in both in vivo and in vitro breast cancer models that S100A4 controls the peroxisome proliferator-activated receptor γ (PPAR-γ)-dependent/fatty acid oxidation pathway, which is the driver of the pro-tumor M2-like polarization of tumor-associated macrophages (TAMs) [194]. Interestingly, S100A4 was predominantly expressed in CD206 + TAMs with a prototype phenotype. S100A4 expression in macrophages induced the immunosuppressive phenotype of M2-like macrophages, including the key markers CD206, arginase-1 (Arg-1), programmed death ligand 1 (PD-L1), and transforming growth factor beta (TGF-β). In 2023, Ding and his colleagues used single-cell transcriptome analysis of databases to establish six key genes (CD53, TGFBI, S100A4, pyruvate kinase M, LSP1, SPP1) associated with macrophage polarization, including S100A4 [195]. In 2023, Kazakova et al. performed a study of human colorectal cancer tissues and found S100A4 expression in the stroma, and in partial TAMs correlated with macrophage infiltration [196]. In 2024, Coulton et al. performed a comprehensive analysis of TAM signatures in 17 human tumor types and confirmed the presence of a subpopulation of TAM expressing S100A4 [197]. In 2025, Huang et al. demonstrated the contribution of S100A4+ alveolar macrophages to the progression of precancerous atypical adenomatous hyperplasia (the precursor to lung adenocarcinoma), including angiogenesis modulated by palmic acid-related metabolic reactions of S100A4+ alveolar macrophages [198].

*Tumor microenvironment: T cells.* Research by Grum-Schwensen revealed that a process involving S100A4-mediated secretion of CCL24 and g-CSF by tumor cells plays a key role in the infiltration of T cells into the premetastatic niche [192]. This process also disrupts the Th1/Th2 balance, favoring the Th2 pro-tumorigenic phenotype. This, in turn, contributes to the development of metastases. Thus, the authors suggest that blocking S100A4-dependent chemotaxis is a promising therapeutic approach (see the chapter on therapy). S100A4 expression was found in the T-cell component of the tumor microenvironment, including regulatory T-cells and exhausted T8 cells. S100A4 has an immunomodulatory effect on T lymphocytes and can form an immunosuppressive microenvironment by inducing expression of IL-10 in T cells [199].

*The tumor microenvironment: natural killer cells.* The expression patterns of S100A4 and S100A6 correlated with the developmental stages of cytotoxic natural killer (NK) cell subsets. Both proteins were recruited into the F-actin-rich NK immune synapse, following NK cell activation [200,201,202]. NK cell subsets that are more mature and possess higher cytotoxic potential also show the highest activation of LFA-1, which correlated with the expression of protein S100A4 [203]. In 2024, Rebuffet et al. employed S100A4 as a marker for three prominent NK cell subsets (NK1, NK2, and NK3) in healthy human blood and demonstrated that expression of S100A4 was exclusive to the NK3 subset [204].

*The tumor microenvironment: myeloid-derived suppressor cells.* Li et al. demonstrated the critical role of S100A4 in preventing the apoptosis of myeloid-derived suppressor cells (MDSCs) in vivo via TLR4-ERK1/2 signaling [205]. In 2022, Abdelfattah et al. identified S100A4 as a regulator of immune suppressive T and myeloid cells in glioblastoma and demonstrated that deleting S100a4 in non-cancer cells is sufficient to reprogram the immune landscape and significantly improve survival [206]. Elevated S100A4 positively correlates with MDSCs and several immune checkpoints [207]. S100A4 activates the GP130/JAK2/STAT3 pathway to promote MDSC accumulation and functions in acute myeloid leukemia via interacting with GP130.

*The tumor microenvironment: B-cells.* Zamzam showed that S100A4 is predominantly expressed in T-cells, rather than B-cells, under normal conditions [208]. However, S100A4 can be activated in B-cell lymphomas, where it contributes to their ability to migrate and invade other tissues. Further investigation is required to determine any specific impact on B-cells and their anti-tumor functions, such as antigen presentation and antibody production.

*The tumor microenvironment: dendritic cells.* S100A4 promotes adaptive immunity through dendritic cells (DCs) [23]. Sun et al. (2017) demonstrated that mice lacking S100A4 have a defective humoral and cellular immune response to mucosal (sublingual) immunization with a model protein antigen (ovalbumin, OVA) given together with the strong mucosal adjuvant cholera toxin, and that this impairment is due to defective adjuvant-stimulated antigen presentation by antigen-presenting cells [209]. Adoptive transfer of WT DCs, but not S100a4-/- DCs, into S100a4-/- mice one day prior to immunization with OVA could restore humoral and cellular immune responses, implicating the importance of S100A4 in regulating adaptive immunity through DCs. Recent studies have demonstrated that extracellular S100A4 signaling through RAGE and TLR4 can enhance the activation of DCs by increasing the expression of cytokines that are critical for adaptive immunity, such as IL-2 and IL-6, as well as co-stimulatory molecules, including CD80, CD86, and CD40 in vitro [210]. S100A4 has also been demonstrated to be involved in the tolerogenic function of DCs. The present study demonstrated that human monocyte-derived DCs pulsed with soluble S100A4 increased the proliferation of CD4+ CD25+ FOXP3+ regulatory T cells in vitro [211].

Therefore, as demonstrated by the analysis, the S100A4 protein is a pivotal regulator of oncogenesis and should thus be regarded as a significant prognostic marker for numerous cancerous diseases. The multifaceted role of S100A4 in tumor progression is realized through a complex network of intracellular and extracellular signaling pathways. S100A4 fulfils a pivotal function as both a marker protein for the classification of cancer-associated fibroblasts and a factor in their activation.

## 4. The Importance of S100A4 in the Formation of Fibrotic Changes in Inflammatory Processes

Reviews have extensively covered the significant role of S100A4 in the formation of stroma and concomitant inflammatory reactions in pathological processes [23,212,213]. According to the results of many years of observation, S100A4 was nominated as a marker of fibrosis and matrix remodeling in inflammatory processes. Typically, plasma S100A4 levels increase in patients with acute inflammation, as does S100A4 expression in resident fibroblasts.

*Hypertrophic heart.* In 2013, Yodo Tamaki et al. demonstrated an in vitro regulatory effect of S100A4 on collagen expression and fibroblast proliferation in a cardiac fibroblast model, mediated by interaction and modulation of p53 functions [213]. Furthermore, the critical role of S100A4 in the formation of interstitial fibrosis, proliferation of myofibroblasts, expression of collagens, and profibrotic cytokine was confirmed in the transverse aortic constraints model in S100A4 knockout mice. 

*Rheumatoid arthritis (RA).* An increase in S100A4 expression was shown in activated synovial fibroblasts with the progression of the disease, along with high levels of S100A4 in the plasma and synovial fluid of RA patients [214]. In addition to fibroblasts, immune and vascular cells in synovial tissue expressed S100A4 [215]. The number of S100A4 was significantly higher in the serum of RA patients and correlated with a poor prognosis. S100A4 interacts with p53 tumor suppressor in RA synovial fibroblast-like cells and affects downstream targets, including Bcl-2, p21 (WAF), Hdm-2, and MMPs [216]. S100A4 has also been demonstrated to upregulate MMP-3 in rheumatoid arthritis synovial fluid, along with MMP-1, MMP-9, and MMP-13 [217]. It has been shown that human articular chondrocytes produce S100A4, and that S100A4 can stimulate the production of MMP-13 by articular chondrocytes via a receptor for RAGE-mediated signaling [140]. High S100A4 levels correlate with RA drug resistance to infliximab therapy, a high rate of anti-infliximab antibodies. These findings suggest that S100A4 could serve as a prognostic marker and a marker of drug resistance for RA [218].

Increased expression of S100A4 in cartilage is a hallmark of inflammatory diseases such as rheumatoid arthritis and osteoarthritis [217]. Matrix remodeling during inflammation is closely related to the activity of metalloproteases, which are known for their proteolytic activity against matrix components and their role in invasion processes. The increased expression of MMP-13 metalloproteases is triggered by interaction with RAGE receptors in chondrocytes. More detailed studies on the initiation of metalloproteases overexpression have shown that proinflammatory cytokine Interleukin-1β plays a leading role in the movement of S100A4 into the cell nucleus, where MMP13 expression is regulated at the transcriptional level. Interestingly, post-translational modification of the simulation of S100A4, involving the addition of a SUMO peptide to a lysine residue, is necessary for the transition of S100A4 into the core [219].

*Liver fibrosis.* Researchers studying S100A4 expression in liver fibrosis found that the source of S100A4 is macrophages that cause the overexpression of alpha-smooth muscle actin through c-Myb in HSCs [220,221].

*Lung fibrosis.* Two independent groups of researchers have shown that S100A4 is expressed by S100A4+ CD11b+ F4/80+ macrophages under conditions of pulmonary fibrosis [222,223]. These cells secrete S100A4 into the extracellular space, stimulating the proliferation and activation of pulmonary fibroblasts by upregulation of α-SMA and type I collagen. The expression of sphingosine-1-phosphate also increases [224]. In the bleomycin model of induced pulmonary fibrosis in vivo, an increase in the FSP1(+) fibroblasts caused collagen deposition. A high S100A4 content was also found in the bronchoalveolar lavage of patients with idiopathic pulmonary fibrosis. S100A4 has been described as a mediator of fibrogenicity in mesenchymal progenitor cells, which have been defined as precursors for IPF-mediating myofibroblasts [225].

*Renal fibrosis.* In 2022, Wen et al. showed the impact of S100A4 on the development of fibrotic changes in the kidneys, which involved the TGF-β1/Smad signaling pathway, promoting fibroblast activation [82]. Mechanistically, S100A4 binds to Smad3, increasing Smad3/Smad4 complex formation and nuclear translocation, which leads to increased expression of profibrotic proteins. The authors suggest that S100A4 is an attractive target for therapies directed against chronic kidney disease.

The importance of S100A4 for the development of fibrosis in inflammatory diseases is also supported by the effectiveness of monoclonal antibodies in preventing fibrotic changes. In 2024, Trinh-Minh et al. showed that an anti-S100A4 monoclonal antibody can reduce bleomycin-induced skin fibrosis by modulating the STAT3, AMP-activated protein kinase, and calsequestrin-1 signaling pathways [226]. In the same year, Švec et al. obtained similar data on the antifibrotic and anti-inflammatory effects of the 6B12 monoclonal antibody on bleomycin-induced dermal fibrosis [227].

The following proteins are associated with the action of S100A4 in the formation of fibrosis and the initiation of the inflammatory process:-Metalloproteases: S100A4 increases the expression of various metalloproteases in different types of cells, including smooth muscle cells (MMPs 1,2,3,9) and articular chondrocytes (MMP-13). Osteoarthritis synovial fibroblasts treated with S100A4 oligomer can induce the expression and release of MMP-3 and other MMPs (MMP-1, MMP-9, and MMP-13), thereby promoting extracellular matrix remodeling;-Interaction with the RAGE receptor allows S100A4 to weaken autophagy, contributing to the development of fibrosis and angiogenesis.

Consequently, the S100A4 protein has been identified as a pivotal mediator of fibrotic degeneration in a range of chronic inflammatory diseases and oncological processes. Its increased expression by fibroblasts and immune cells directly correlates with the activation of myofibroblasts, excessive deposition of various types of collagen, and pathological remodeling of the extracellular matrix.

## 5. Therapeutic Approaches to Targeting S100A4

A possible blockage of S100A4 functions may be associated with effects on various signaling pathways that affect tumor cell invasiveness. These pathways include S100A4/myosin-IIA, β-catenin, RAGE, and NFκB, and TP53 signaling.

*Inhibitors of myosin-II associated with S100A4.* Trifluoperazine was identified as an inhibitor that disrupts the S100A4/myosin-IIA interaction by sequestering S100A4 via small-molecule-induced oligomerization, thereby blocking the exercise of S100A4 function, including the mediation of carcinoma cell motility via interaction with myosin-IIA [228]. The identification of phenothiazines as inhibitors of myosin-II-associated S100A4 function was performed by screening an FDA-approved drug library with the fluorescent biosensor (Mero-S100A4), which recognizes the Ca^2+^-bound, activated form of S100A4 [229]. This biosensor was created by Anne R. Bresnick’s group in 2008.

*Niclosamide.* In 2011, Sack et al. conducted a large-scale high-throughput screening of 1280 pharmacologically active compounds using a colon cancer cell-based model that expressed a S100A4 promoter-driven luciferase reporter gene construct [230]. They found that niclosamide, an antihelminthic agent, could reduce S100A4 mRNA and protein expression and block S100A4-mediated functions, including tumor cell migration, invasion, and proliferation. Niclosamide has also been shown to reduce metastasis formation in a mouse model of colon cancer. Its mechanism of action is associated with modulating beta-catenin—mediated activation of S100A4 transcription. New studies in 2022 have shown successful in vivo synergy in reducing metastasis and tumor cell migration in vitro by targeting the MACC1-β-catenin-S100A4 pathway by statins (MACC1) and niclosamide (S100A4) [231]. Metastasis-associated in colorectal cancer-1 (MACC1) enhances the interaction between β-catenin and TCF4. This leads to increased S100A4 transcription. Blocking this axis helps reduce colorectal cancer metastasis. In 2008, an additional mechanism of action of niclosamide was discovered [232]. Extracellular S100A4 specifically activates NF-κB in a subset of human cancer cell lines via the classical NF-κB activation pathway. Components of this signaling pathway include ephrin-A1 and optineurin, which may mediate the effects of S100A4. In 2016, Stewart et al. demonstrated the importance of the S100A4/NF-κB/MMP9 pathway in lung cancer cell invasion and the possibility of niclosamide to inhibit it [233]. In 2018, Liu et al. showed that niclosamide reduced breast cancer metastasis in vivo and mitigated the effects of CAFs on tumor cell migration [182]. In 2022, Treese et al. demonstrated that blocking S100A4 reduces the migration of adenocarcinoma of the gastroesophageal junction and stomach (AGE/S) cells when treated with niclosamide [74]. In 2024, the localization of S100A4 in benign prostatic hyperplasia cells was established, as was the stroma caused by niclosamide. Knockout of S100A4 resulted in apoptosis and decreased tissue fibrosis markers [136]. Thus, niclosamide is an effective anticancer drug that is promising for clinical studies.

In 2016, the Phase II NIKOLO trial protocol was published to assess the safety and efficacy of orally applied niclosamide in patients with metachronous or synchronous metastases of a colorectal cancer (NCT02519582) [234]. Patients are currently being recruited.

*Sulindac.* In 2011, it was shown that the nonsteroidal anti-inflammatory drug Sulindac sulfide decreased colon cancer metastasis by intervening in β-catenin/TCF/S100A4 signaling [235]. Sulindac was able to repress the expression and nuclear accumulation of β-catenin with less binding to TCF. This reduced S100A4 promoter activity and expression, leading to the inhibition of cancer cell migration and invasion. 

*Small RAGE antagonist peptide (RAP).* In 2012, Arumugam et al. created a small RAGE antagonist peptide (RAP) that inhibits the interaction of S100P, S100A4, and HMGB-1 with RAGE at micromolar concentrations [236]. This prevented RAGE-mediated activation of NFκB in pancreatic and glioma cancer cells in vitro and in vivo, leading to decreased tumor growth and metastasis in vivo.

*PROTAC.* In 2023, Ismail et al. (the Rudland team) developed a proteolysis targeting chimera (PROTAC) consisting of the S100A4 inhibitor US-10113, which is covalently linked to thalidomide. This chimera successfully degraded S100A4 in rat (IC50, 8 nM) and human TNBC (IC50 3.2 nM) cell lines, as well as inhibiting cell migration (IC50 3.5 nM) [237].

*Amlexanox.* In 2022, Christian Bailly and colleagues described the antitumor effect of the drug Amlexanox (azoxanthone; used to treat mouth aphthous ulcers) in vivo against breast, colon, lung, blood, and gastric cancers. The mechanism of action was mainly directed at inhibitor-kappaB kinase epsilon (IKK-ε/TBK1), but it also included S100A4 [238].

*Monoclonal antibodies.* Ganaie AA et al. tested monoclonal antibody 6B12 in vivo in a TRAMP-C2 syngeneic mouse model of prostate cancer. They found decreased tumor growth and osteoblastic demineralization of bone-derived MSCs, which makes this antibody promising for the clinic [239]. The monoclonal antibody 6B12 was shown to restore the Th1/Th2 balance and reduce the chemotaxis of immune cells in the primary tumor and metastasis area. This, in turn, reduced tumor growth and metastasis in a model of spontaneous breast cancer in vivo [192]. The monoclonal antibody S1004-11 effectively suppresses breast cancer metastasis in two different mouse models [50]. Combined inhibition of S100A4 and TIGIT suppresses late-stage breast cancer metastasis to the lung by activating T and NK cells.

Notwithstanding the execution of a Phase 1 clinical trial (NCT05965089) for patients suffering from mild to moderate chronic plaque psoriasis, it is imperative that further safety studies are conducted for the potential utilization of antibodies in clinical practice. Previous studies have also shown that 6B12 reduces fibrotic and inflammatory changes in vivo, as well as decreases profibrotic and proinflammatory markers, as determined by bulk RNA sequencing (RNAseq) when treated with human systemic sclerosis-derived fibroblasts. These data indicate the potential of using the antibody to prevent fibrotic changes in the tumor environment.

*siRNA vs. S100A4*. In 2014, Ochiya et al. found that intra-tumor administration of the S100A4 siRNA in a human prostate cancer xenograft model reduced tumor vascularity and tumor growth [240].

*Pentamidine.* S100A4 has previously been shown to destabilize p53 by provoking an increase in tumor cell proliferation [241,242]. Pentamidine, via a competitive binding mechanism, counteracts S100A4-p53 and increases the level of p53, reducing the proliferation of MCF7 and ZR-75-1 breast cancer cells.

Additional data on the side effects of the drugs targeting the S100A4 protein can be found in Appendix A.

## 6. Discussion and Future Directions

An unfavorable cancer prognosis is associated with metastatic progression and resistance to drug therapy [243]. In this regard, there is still a need to search for and develop a diagnostic platform that evaluates prognostic biomarkers expressed by tumor and microenvironment cells. Most studies support the prognostic value of determining S100A4 expression in primary tumors for predicting a poor cancer prognosis in patients [244,245,246].

S100A4 is overexpressed in tumors compared to normal tissues, and it is involved in many signaling pathways [24]. It promotes proliferation, metastasis, the formation of an immunosuppressive tumor microenvironment, and fibrotic changes that impede drug infiltration and cause resistance to therapy [24,247]. Therefore, the immunocytochemical determination of S100A4 expression could be recommended for inclusion in the prognostic biomarker panel to predict patients’ outcome [248]. Additionally, S100A4-containing exosomes can be detected in the plasma of patients with liver cancer. Further research on exosomes could demonstrate their potential for liquid diagnosis of other forms of cancer [76].

From the perspective of identifying novel therapeutic agents to combat metastatic stages of cancer, S100A4 emerges as a promising target protein. This assertion is substantiated by the documented correlation between its elevated expression by tumor cells and diminished patient survival, a criterion widely regarded as the gold standard for target selection. A comprehensive analysis of numerous publications on the mechanisms of action of S100A4 has identified several primary directions for preventing the development of metastasis and warming up of the cold microenvironment associated with signaling pathways mediating S100A4 functions.

First of all, the objective is to regulate S100A4 expression at the transcriptional level by targeting the Wnt/β-catenin signaling pathway. A series of small-molecule screenings has been carried out with the objective of identifying compounds capable of impeding β-catenin-mediated activation of S100A4 expression. These screenings have demonstrated efficacy in curtailing colorectal cancer metastasis in preclinical studies, with calcimycin, sulindac, and niclosamide being notable examples. Hence, subsequent studies can be directed towards evaluating the efficacy of this therapy in other nosologies associated with Wnt/β-catenin signaling. Furthermore, S100A4 can be utilized as a biomarker to guide the prescription of this therapy. However, when selecting nosological forms, it is imperative to meticulously consider the documented neuroprotective effect of S100A4, while concurrently monitoring for the emergence of potential neurological complications.

In the future, the use of combinatorial therapy aimed at blocking several axes of transcription-regulating S100A4, for example, combined inhibition of MACC1 statins and β-catenin niclosamide, may be considered [231]. It is noteworthy that S100A4 plays a regulatory role in the transcription of ERBB2, thereby offering an additional opportunity to reduce S100A4 levels through the use of ERBB2 inhibitors. Concurrently, the administration of multiple medications has the potential to exacerbate adverse effects, and the identification of the most effective combination of drugs remains an area of ongoing research.

The next effective direction for the development of antitumor agents is to block the binding of the extracellular form of S100A4 to RAGE receptors. Extensive research has demonstrated the significance of the MAPK/Erk signaling S100A4/RAGE/cascade in regulating tumor cell migration and invasion, as well as profibrotic alterations within the microenvironment. Inhibiting the S100A4/RAGE interaction will also prevent the formation of a prometastatic niche in the bones. In this regard, the prospect of obtaining monoclonal antibodies capable of binding and neutralizing the effects of the extracellular form of S100A4 is a promising venue for further research. Indeed, the encouraging effects have been observed with the administration of 6B12 monoclonal antibody, which has been demonstrated to impede the growth and metastasis of breast cancer. This finding lends credence to the continued utilization of the antibody in ongoing clinical trials.

In addition to binding the extracellular form of S100A4, it is possible to reduce the migration and invasion of tumor cells by inhibiting the interaction of intracellular S100A4 with proteins of various signaling pathways involved in the progression of tumor diseases.

Therefore, it can be posited that the use of phenothiazines may serve to reduce the mobility of tumor cells, a process that is facilitated by the interaction between S100A4 and NMIIA [229]. The following intracellular signaling pathways are important for tumorigenesis: S100A4/MTA1/NM, which is responsible for angiogenesis; TGFbeta/S100A4, which promotes tumor cell migration and immunosuppressive effects of CAFs.

Of particular interest is also the suppression of the S100A4 effect in maintaining tumor stem cells, as CSCs are responsible for invasion and chemoresistance. The suppression of these pathways is a multifactorial strategy that aims to reduce metastasis. In this regard, the development of the targeted degradation technology known as PROTAC shows promise. In vitro studies demonstrated the ability of PROTAC to degrade intracellular S100A4 at nanomolar concentrations [237]. Future advancements in this field will entail the in vivo testing of PROTAC in metastatic models and the toxicological studies. Additionally, a potential combination with drugs already utilized in the clinical setting, including TGFbeta inhibitors, is under consideration.

In light of the substantial body of evidence regarding the impact of S100A4 expression on microenvironment cells, the utilization of inhibitors emerges as a plausible strategy to transition the “cold” microenvironment into a “hot” one. These factors include the polarization of Th1/Th2 lymphocytes, the polarization of macrophages (M1/M2), and the polarization of MDSCs. The inhibition of signals within the microenvironment with inhibitors has been demonstrated to reduce immunosuppression and to direct an immune response to tumor cells.

The phenomenon of matrix stiffness has been identified as a primary factor contributing to the observed radio and chemical resistance. S100A4 is a critical element in the progression of fibrosis in inflammatory and oncological diseases. Blocking enables a reduction in the barrier to drug entry, a strategy that can be employed in the development of a combination therapy and to achieve synergy with another drug. This approach facilitates the reduction of dosage while circumventing the occurrence of adverse effects.

Additionally, the observation of S100A4/STAT3-dependent overexpression of the known PD-L1 immunotherapy target is noteworthy. The combination of inhibitors targeting these factors with a synergistic warming effect on the tumor microenvironment appears to be promising, as it is expected to enhance the sensitivity of patients to therapy with immune checkpoint inhibitors.

According to the classification proposed by Costa et al., there is a distinction between more aggressive phenotypes of CAF-S1 and CAF-S4 and non-activated, less aggressive S2-S3 CAFs [182]. The latter are closer in phenotype to normal fibroblasts. Our recent study supports the observations of numerous colleagues regarding the high plasticity of CAFs and their capacity to transition between different subtypes, forming hybrid forms in some scenarios [249]. The subject of modern discussion is the possible mechanisms of initiation of the transition of normal fibroblasts to CAFs (MMT transition), as well as the further development of CAF forms associated with metastasis and immunosuppression. Based on the comprehensive analysis of the extant literature, we may suggest that activation of FSP1 transcription, which is mediated by the Wnt/beta-catenin signaling pathway, plays an important role in the MMT transition of normal stromal cells to CAF-S3 (Figure 5). Furthermore, CAF-S3 may transit to aggressive CAF phenotypes through the activation of RAGE downstream pathways, including MAPK/ERK and Dia-1 signaling [60,104]. The detailed mechanism of CAF transition and the pathways by which they maintain drug resistance are the subject of our further studies, along with the search for inhibitors that can reverse the more malignant phenotype to the non-activated state to increase drug sensitivity and improve patient prognosis.

In our recent studies, S100A4 has been utilized for the successful typing of fibroblasts, thereby enabling the characterization of a unique new collection of CAFs [249]. Further research in our laboratory will be related to the study of the role of S100A4-expressing fibroblasts in the tumor microenvironment and the possibility of inhibiting S100A4 to overcome drug resistance.

## 7. Conclusions

A review of recent literature reveals that S100A4 fulfils several pivotal functions in the progression of cancer, fibrosis, and autoimmune diseases, and in the immune system. S100A4 belongs to the extensive group of S100 proteins, which exhibit a wide array of intracellular and extracellular functions that are subject to variation depending on the specific cellular context [136]. While S100A4 has long been implicated mainly in tumorigenesis and metastasis, mounting evidence shows that S100A4 is a key player in promoting pro-inflammatory phenotypes and organ pro-fibrotic pathways in the liver, kidney, lung, heart, tendons, and synovial tissues [30].

Detection of S100A4 expression becomes a promising candidate biomarker in cancer early diagnosis and prediction of cancer metastasis, and therefore, S100A4 may be a therapeutic target. The strategies for therapeutically targeting S100A4 in the treatment of human cancers have been evaluated in preclinical studies, including RNAi-based knockdown, S100A4 signaling inhibitors, S100A4-specific antibodies, drug/peptide/small molecule-based interference of S100A4-protein interactions, and other inhibitors [24].

As such, this review comprehensively outlines the role of S100A4 in cancer initiation, advancement, and spread, detailing the associated molecular pathways and experimental strategies to modulate S100A4 expression.

## Figures and Tables

**Figure 1 ijms-26-09370-f001:**
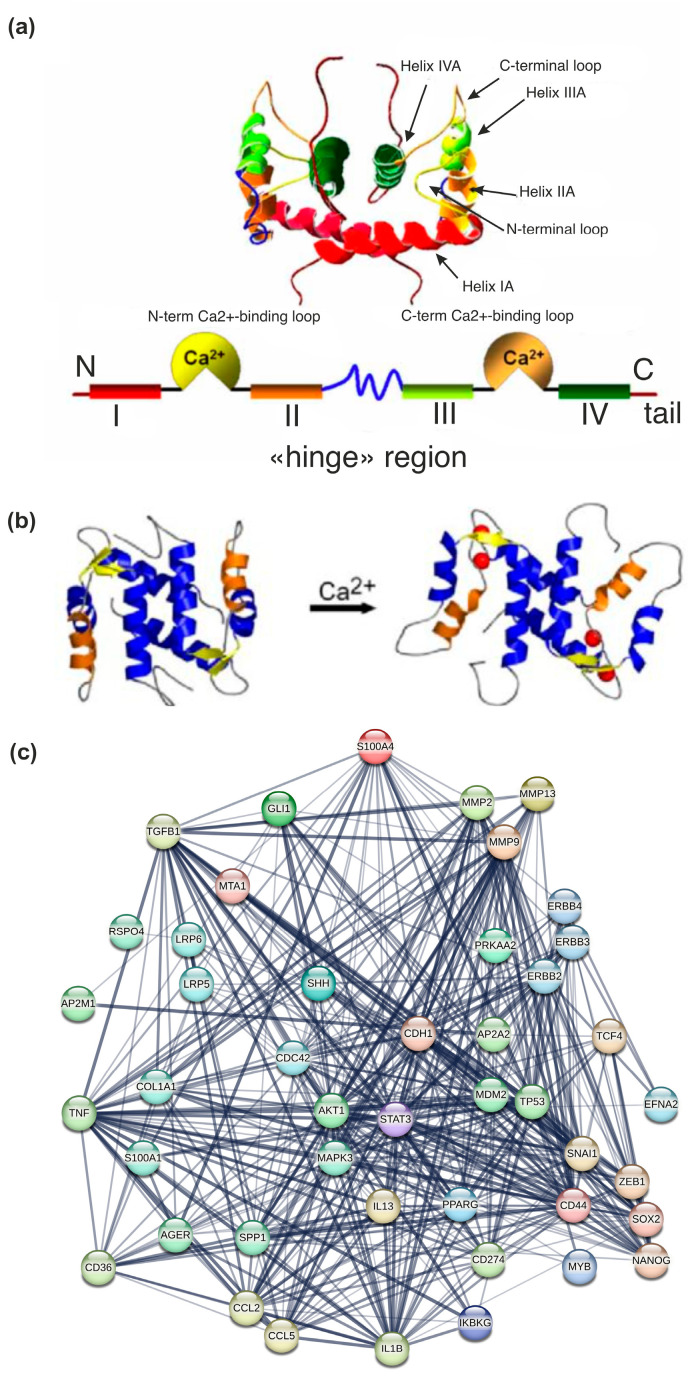
(**a**) 3D solution structure of the apo-S100A4 protein and structural domains of the S100A4 protein in linear mode (I–IV is helical regions). Adopted from [36] (**b**) S100A4 undergoes a calcium-dependent conformational rearrangement that exposes the protein target binding cleft [37]. (**c**) STRING: proposed network of interacting proteins for S100A4 [38]. Additional data are shown in Appendix A.

**Figure 2 ijms-26-09370-f002:**
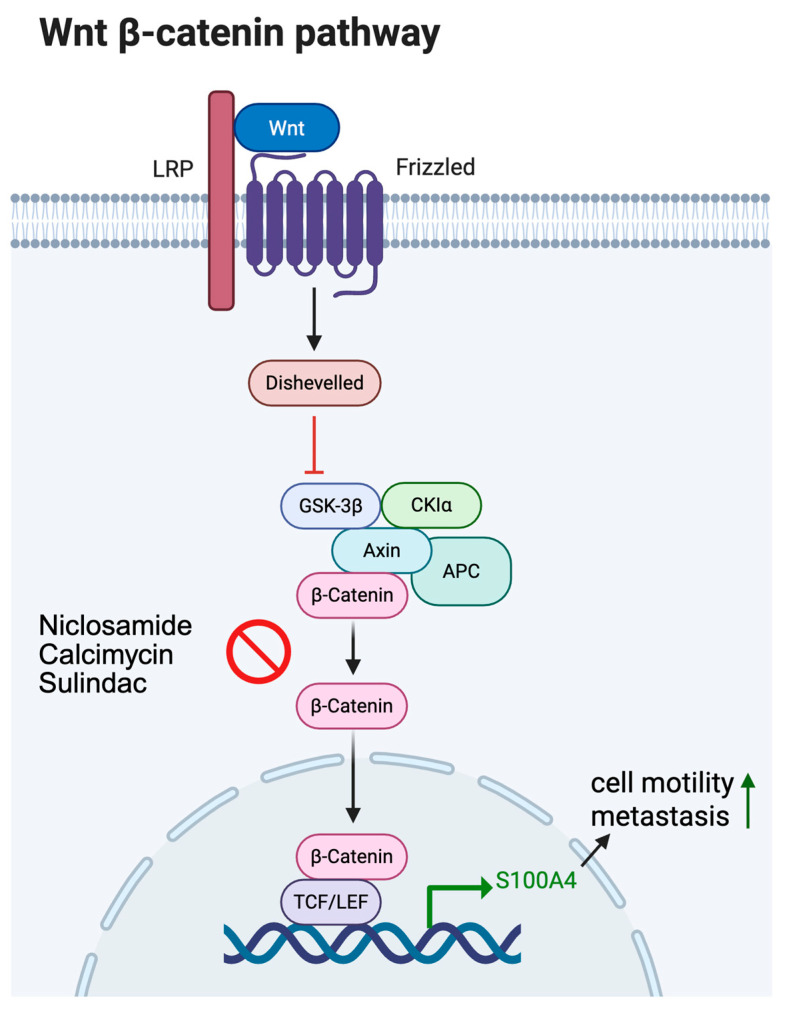
Wnt/β-catenin signaling regulates S100A4 at the transcription level. The phosphorylation of β-catenin via a destruction complex serves as a signal for its subsequent degradation by the proteasome. During tumorigenesis, β-catenin is not subject to degradation and accumulates within the cytoplasm. Further, β-catenin migrates to the nucleus, where it binds to the heterodimeric β-catenin/T-cell factor complex, resulting in the activation of S100A4 transcription.

**Figure 3 ijms-26-09370-f003:**
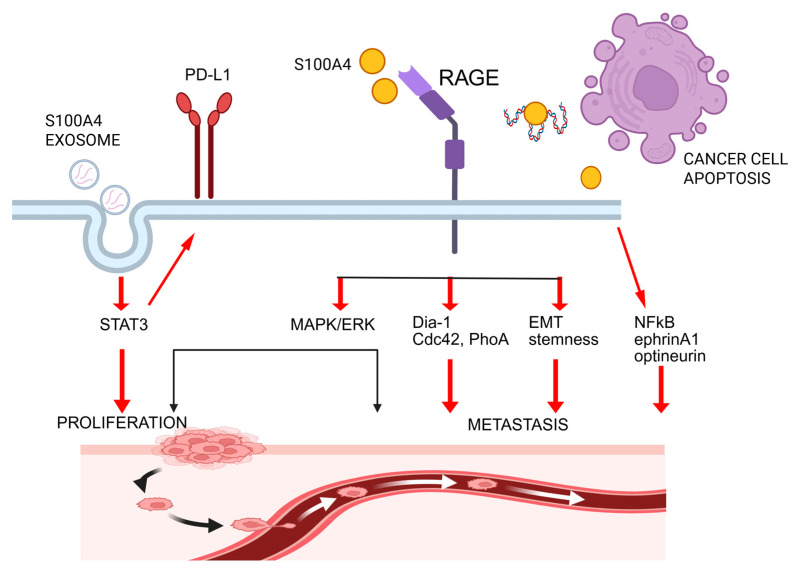
Downstream cascade of extracellular S100A4. S100A4 triggers EMT, stemness, tumor proliferation, and metastasis via (1) S100A4/RAGE/downstream signaling and alternative (2) NF-κB signaling pathway. Vesicular S100A4 activates STAT3 and increases the overexpression of PD-L1. Apoptotic cancer cells produce DNA/S100A4 complexes with pro-carcinogenic effects.

**Figure 4 ijms-26-09370-f004:**
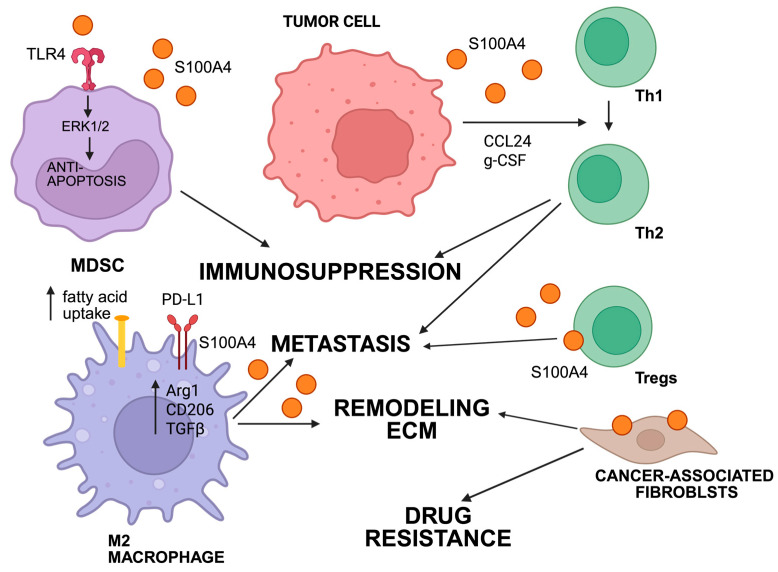
Impact of S100A4 on tumor microenvironment. S100A4 causes immunosuppression, metastasis and fibrosis due to impact on cellular component of TME: (1) S100A4/TLR4/ERK 1/2 signaling provides anti—apoptotic effect on immunosuppressive MDSCs; (2) Th1/Th2 transition toward immunosuppressive Th2 phenotype; (3) expression of S100A4 by immunosuppressive Tregs; (4) polarization of macrophages toward pro carcinogenic M2 macrophages type with overexpression of immunosuppression markers; (5) activation aggressive phenotype of CAFs.

**Figure 5 ijms-26-09370-f005:**
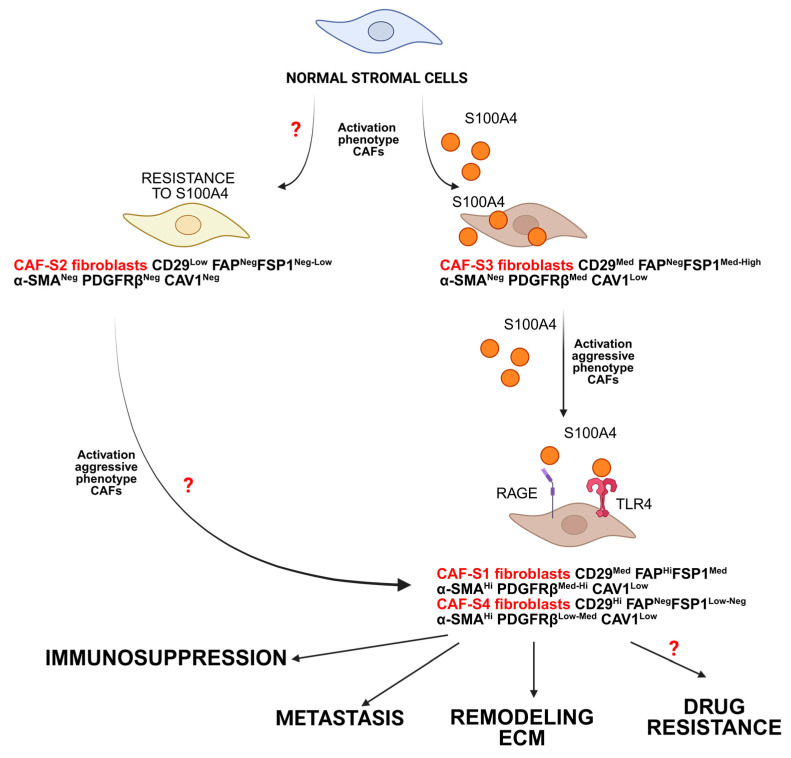
Possible mechanism of normal stromal cell transitions in CAFs and transition of CAFs subtypes with formation of CAF aggressive phenotypes. S100A4 resistance possibly supports the transition of normal stromal cells to the CAF-S2 subtype, while S100A4-sensitive stromal cells transit to the CAF-S3 subtype. CAF-S3 may transit to aggressive CAF phenotypes through the activation of RAGE downstream pathways supporting ECM remodeling and metastasis development. The mechanism (symbol “?”) of CAF—mediated development of drug resistance and transition of normal stromal cells to the CAF-S2 subtype remains the subject of further study.

## Data Availability

The data presented in this study are available on request from the corresponding author.

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
