# Peer review of "S100A4/FSP1: A Prognostic Marker and a Promising Target for Antitumor Therapy"

_ijms, 2025, doi:10.3390/ijms26199370_

Round 1

Reviewer 1 Report

Comments and Suggestions for Authors

This review of the literature regarding the role that the S100A4 protein plays in the progression of cancer, fibrosis, autoimmune diseases, and in the immune system as a whole is enriching and extremely important for studies that are being developed, trying to show the efficiency and therapeutic efficacy of this protein, especially in relation to calcium deficiency. Therefore, I believe that this review can help enhance and optimize many studies related to the S100A4 protein, calcium, and diseases such as cancer, fibrosis, and autoimmune diseases. Furthermore, the promise is that it can be used as a biomarker for early diagnosis of cancer and prediction of metastases through various molecular pathways that present resistance to drugs, making it possible to reverse the malignant phenotype through the action of inhibitors found in the tumor microenvironment that can increase sensitivity to drugs by inhibiting the S100A4 protein and, therefore, improving the patient's prognosis. 

Congratulations for authors and I hope it is promising for many others.

Comments on the Quality of English Language

In my opinion, the English used is clear, objective, and accessible to academic, scientific, medical, and healthcare audiences as a whole. Furthermore, the grammar used in the third person was appropriate, and the technical terms were used appropriately.

However, I must emphasize the consistency throughout the text, which enhances the article's overall coherence.

Author Response

Reviewer 1.

We thank the reviewer for their positive feedback on our work.

In the current version of the manuscript, we corrected minor spelling and typo errors. Current version of the manuscript was proofed by English speaker.

Reviewer 2 Report

Comments and Suggestions for Authors

Maria Bogachek et al. provided an in-depth analysis of the role of S100A4 protein in tumor progression, fibrosis, and the immune microenvironment, as well as its potential as a therapeutic target. Overall, this manuscript is well-written and well-organized. However, I think this manuscript needs minor revision and is not suitable for publishing in International Journal of Molecular Sciences in its current form. Here are some comments of this manuscript for authors.

  1. The article does not mention the limitations of the current S100A4 research (e.g., specificity, safety).
  2. Chapter 3 “S100A4: Significance for Oncogenesis and Prognosis of Cancer” and Chapter 4 “The Importance of S100A4 in the Formation of Fibrotic Changes in 753 Inflammatory Processes” are heavy reading and lack summarization. It is recommended to create tables organized by “disease type” to summarize expression levels, prognostic significance, and mechanism pathways.
  3. Please standardize the formatting of “et al.” throughout the text to either upright or italic type. Currently, the formatting is inconsistent.
  4. Was S100A4 exploited as a potential therapeutic target or biomarker in clinical practice?
  5. In addition to cancer and fibrosis, does S100A4 also influence other types of diseases?
  6. Figure 5 is not clear enough. Please redraw it.
  7. Please carefully check whether the reference format is consistent.

Author Response

Reviewer 2.

Comments for Reviewer 2:

Point 1. The article does not mention the limitations of the current S100A4 research (e.g., specificity, safety).

Response 1: Regarding the limitations of the studies, attention should be paid to the issue of redundancy S100A4. Indeed, there are other members of the S100 family that affect carcinogenesis reviewed in https://doi.org/10.2147/CMAR.S508047. Thus, blocking S100A4 should take into account the possibility of activating additional pathways mediated by homologous proteins. Regarding safety issues we can mention that currently most S100A4 inhibitors are in preclinical efficacy studies and safety issues are still in progress and a question of the future research.

However, it could be noted that PROTAC was able to inhibit S100A4 action in nanomolar amounts (in the rat (IC50, 8 nM) and human TNBC (IC50, 3.2 nM) in vitro PMCID: PMC10377353  PMID: 37509135, giving hopes for potential application of low doses of S100A4 in vivo, reducing potential toxic effects.

Also of interest is the retargeting of the well-established anthelmintic drug Niclosamide, which is capable of inhibiting S100A4 and has a generally favorable oral safety profile, with the most common side effects being mild to moderate gastrointestinal issues.

It was reviewed in Chapter 5 (Therapeutic Approaches to Targeting S100A4) that in 2016, the Phase II NIKOLO trial protocol was published to assess the safety and efficacy of orally applied Niclosamide in patients with metachronous or synchronous metastases of a colorectal cancer (NCT02519582) [229]. https://clinicaltrials.gov/study/NCT02519582 and the requirement of the patients is still needed for the detailed studies of safety.

Also, a more comprehensive understanding of S100A4's extracellular functions is required for its effective targeting in therapies.

We also emphasized the need for safety studies on the use of monoclonal antibodies in the section Therapeutic Approaches to Targeting S100A4, Monoclonal antibodies.

Point 2. Chapter 3 “S100A4: Significance for Oncogenesis and Prognosis of Cancer” and Chapter 4 “The Importance of S100A4 in the Formation of Fibrotic Changes in 753 Inflammatory Processes” are heavy reading and lack summarization. It is recommended to create tables organized by “disease type” to summarize expression levels, prognostic significance, and mechanism pathways.

Response 2:

We have added summarizations after Chapter 3 and 4, and Tables S2 and S3 (included in Supplementary) (See line 863-867).

Point 3. Please standardize the formatting of “et al.” throughout the text to either upright or italic type. Currently, the formatting is inconsistent.

Response 3: Checked, standardized.

Point 4. Was S100A4 exploited as a potential therapeutic target or biomarker in clinical practice?

Response 4: Currently, we observe a transition of the fundamental knowledge accumulated over 30 years about the contribution of S100A4 to the development of cancer to the initial preclinical studies of the efficacy in vitro and in vivo. As mentioned earlier, the most advanced stage is the use of niclosamide, where the recruitment of patients for the Phase II NIKOLO trial was announced. The remaining studies are at the preclinical stage and the results were presented in Section: Therapeutic Approaches to Targeting S100A4.

Point 5. In addition to cancer and fibrosis, does S100A4 also influence other types of diseases?

Response 5: Notably, S100A4 has been implicated in the pathogenesis of various inflammatory and autoimmune diseases, such as rheumatoid arthritis, multiple sclerosis, etc., where it modulates immune cell migration and cytokine production. Furthermore, its involvement in cardiovascular diseases, including atherosclerosis, cardiac hypertrophy, and restenosis, points to a role in vascular remodeling and smooth muscle cell proliferation. Neurological disorders, such as Alzheimer's disease and neuroinflammation, also represent a compelling frontier, as S100A4 is expressed in astrocytes and microglia and may contribute to glial activation and neuronal damage. Therefore, a key future direction will be to systematically explore the mechanisms of S100A4 action in these diverse disease contexts, which could unveil novel therapeutic targets and reposition S100A4 inhibition as a broad-spectrum treatment strategy beyond oncological and fibrotic conditions.

Point 6. Figure 5 is not clear enough. Please redraw it.

Response 6: Checked. Updated.

Point 7. Please carefully check whether the reference format is consistent.

Response 7: Checked.

Reviewer 3 Report

Comments and Suggestions for Authors

Maria Bogachek et al. conducted a comprehensive review of S100A4/FSP1, emphasizing its potential as a promising target for antitumor therapy. This is a well-structured and impactful manuscript that presents a compelling prognostic biomarker in cancer and fibrosis research. However, some more important content should be added.

  • In Introduction elaborate on the role of S100A4 and interplay between fibrosis and cancer.
  • Authors should provide the methods and software used in Figure 1.
  • Please include the table of upregulated and downregulated genes based on the nodes, edges and centrality by using STRING Protein-protein interaction data.
  • The authors ought to clarify the distinct roles of S100A4 in Antigen presenting cells and B- cells.
  • The authors should provide preclinical and clinical data on different solid tumors and their respective treatments.
  • It is prudent to include a table showing the patient’s outcome data or real-world evidence with Progression-Free Survival (PFS).
  • Authors are required to outline the benefits and drawbacks, as well as the side effects, of various treatment regimens available in clinical settings.

Author Response

Reviewer 3.

We are truly grateful to the reviewer for their generous time and exceptionally helpful comments. Their expertise has been instrumental in helping us improve the clarity, rigor, and overall presentation of our manuscript.

Comments to Reviewer 3:

Point 1. In Introduction elaborate on the role of S100A4 and interplay between fibrosis and cancer.

Response 1: We added an additional text to the Introduction (See line 60-73).

Point 2. Authors should provide the methods and software used in Figure 1.

Response 2:

Figure 1c was generated by mapping protein-protein interactions based on available functional and physical association data of proteins with S100A4 from the STRING database (full STRING network). In the STRING web application used to build the interaction network, default settings were applied where the edge thickness indicates the confidence of the data, and clustering was disabled (show network as is). To ensure robust support for the interactions, all available evidence channels were included in the analysis: Textmining, Experiments, Databases, Co-expression, Neighborhood, Gene Fusion, and Co-occurrence. Provided methods are included in the description of Picture.

Point 3. Please include the table of upregulated and downregulated genes based on the nodes, edges and centrality by using STRING Protein-protein interaction data.

Response 3:

As our study was based on a literature-derived protein set rather than an experimental transcriptomic or proteomic dataset, we do not currently have fold-change or expression-level data to classify nodes as upregulated or downregulated. Instead, we have leveraged STRING’s network topology metrics to prioritize proteins by their potential functional importance in the S100A4 interactome. Below, we summarize these metrics:

number of nodes:                              78

number of edges:                               994

average node degree:                        25.5

avg. local clustering coefficient:       0.709

expected number of edges:                405

PPI enrichment p-value:                    < 1.0e-16

STRING indicates that our network has significantly more interactions than expected. This means that proteins in our set “have more interactions among themselves than what would be expected for a random set of proteins of the same size and degree distribution drawn from the genome. Such an enrichment indicates that the proteins are at least partially biologically connected, as a group.”

Because expression values were unavailable, we propose that nodes with the highest degree, betweenness, and closeness centrality be considered as network “hubs” and thus likely critical in S100A4-mediated signaling.

We will include Tables 2 and 3 in supplementary materials of revised manuscript and discuss how these topological features guide future experimental validation, including differential expression analyses and functional assays.

Point 4. The authors ought to clarify the distinct roles of S100A4 in Antigen presenting cells and B- cells.

Response 4:

These descriptions of the distinct roles of S100A4 in Antigen presenting cells and B- cells are included in 3.2. Signaling Pathways that Mediate S100A4 Functions in Tumors and Microenvironment (See Line 761-788).

Point 5. The authors should provide preclinical and clinical data on different solid tumors and their respective treatments.

Response 5: Currently, there is a transition from the knowledge accumulated over two decades about the role of S100A4 in carcinogenesis to preclinical studies of the effectiveness of inhibitors in vitro and in vivo. Thus, the Rudland group, which has studied the relationship between S100A4 expression and an unfavorable prognosis in breast cancer patients since 2000, began initial studies of S100A4 inhibition using the technology of directed degradation of the PROTAC protein. Although toxicity studies are to be performed, initial efficacy data at nanomolar concentrations suggest a safe profile of the new S100A4 inhibitors.

Point 6. It is prudent to include a table showing the patient’s outcome data or real-world evidence with Progression-Free Survival (PFS).

Point 7. Authors are required to outline the benefits and drawbacks, as well as the side effects, of various treatment regimens available in clinical settings.

Response 6 and 7: Due to the fact that clinical trials of drugs targeting S100A4 are not currently being conducted worldwide, it is impossible to provide objective data on patient outcomes and side effects of the drugs.